# Inferior Outcomes of Fludarabine–Cyclophosphamide–Rituximab Chemotherapy in Korean Chronic Lymphocytic Leukemia Patients with Concurrent Thrombocytopenia and Anemia

**DOI:** 10.3390/biomedicines13010194

**Published:** 2025-01-14

**Authors:** Tong-Yoon Kim, Gi-June Min, Young-Woo Jeon, Seung-Ah Yahng, Seok-Goo Cho, Jong-Mi Lee, Myungshin Kim, Ki-Seong Eom

**Affiliations:** 1Department of Hematology, Catholic Hematology Hospital, Yeouido St. Mary’s Hospital, College of Medicine, The Catholic University of Korea, Seoul 07345, Republic of Korea; tyk@catholic.ac.kr (T.-Y.K.); native47@catholic.ac.kr (Y.-W.J.); 2Department of Hematology, Catholic Hematology Hospital, Seoul St. Mary’s Hospital, College of Medicine, The Catholic University of Korea, Seoul 06591, Republic of Korea; beichest@catholic.ac.kr (G.-J.M.); chosg@catholic.ac.kr (S.-G.C.); 3Department of Hematology, Incheon St. Mary’s Hospital, College of Medicine, The Catholic University of Korea, Seoul 21431, Republic of Korea; 4Department of Laboratory Medicine, Seoul St. Mary’s Hospital, College of Medicine, The Catholic University of Korea, Seoul 06591, Republic of Korea; jongmi1226@catholic.ac.kr (J.-M.L.); microkim@catholic.ac.kr (M.K.)

**Keywords:** chronic lymphocytic leukemia, bicytopenia B-cell, chemotherapy, fludarabine, cyclophosphamide, rituximab

## Abstract

**Background/Objectives**: Anti-CD20 monoclonal antibodies combined with alkylator-based chemotherapy enhance survival in chronic lymphocytic leukemia (CLL). However, the risks of infection and bone marrow suppression may mean that new, targeted therapies are more appropriate for some patients than fludarabine–cyclophosphamide–rituximab (FCR). In the Republic of Korea, where insurance limits coverage to novel agents, FCR therapy should be carefully considered for patients with CLL. **Methods**: Using clinical data from 144 FCR-treated patients with CLL, we retrospectively analyzed clinical characteristics impacting survival outcomes, the impact of cytopenia after FCR, and the durable remission status in terms of measurable residual disease (MRD). We compared the impact of bicytopenia with those of other hematologic conditions. **Results**: The 5-year overall survival (OS) and 5-year progression-free survival (PFS) for all patients were 84.4% and 68.3%, respectively. FCR-treated patients in the bicytopenia and *TP53*-positive groups exhibited poor OS and PFS; in particular, the bicytopenia group often experienced prolonged anemia and thrombocytopenia (6–12 months). The responder group achieved sustained remission for a median of 5 years for MRD negativity. **Conclusions**: In bicytopenia, FCR can induce prolonged cytopenia, making it difficult to switch to second-line therapy or complete cycles of chemoimmunotherapy, directly affecting poor survival outcomes. The cautious application of FCR therapy in CLL without bicytopenia or *TP53* positivity can achieve long-term remission.

## 1. Introduction

Chronic lymphocytic leukemia (CLL) is a relatively rare disease that accounts for approximately 2% of all hematologic neoplasms in the Republic of Korea [1,2]. The introduction of anti-CD20 monoclonal antibodies in alkylator-based chemotherapy has substantially improved progression-free survival (PFS) and overall survival (OS) among patients with CLL [3,4]. Fludarabine–cyclophosphamide–rituximab (FCR) therapy is efficacious; however, it is associated with the risk of severe myelosuppression, treatment-related myeloid neoplasms, and infectious complications [5,6]. Hence, combining targeted therapy against Bruton’s tyrosine kinase [7,8], inhibiting B-cell lymphoma 2 [9,10], and/or the anti-CD20 monoclonal antibody has become the frontline treatment [11]. The E1912 trial demonstrated superior OS for patients treated with ibrutinib and rituximab compared with those treated with FCR [8]. Notably, patients who underwent FCR therapy did not show inferior OS rates compared with those treated with ibrutinib, and sustained remissions were achieved through this chemoimmunotherapy [12,13]. Moreover, patients who received FCR and presented with immunoglobulin heavy-chain variable (IGHV) gene mutations could achieve long-term and sustained remissions, indicating that FCR chemotherapy remains a first-line therapy for selected patients. Under the health insurance system in the Republic of Korea, treatment coverage and treatment strategies follow a similar pattern across the country; therefore, FCR is still frequently selected as a first-line therapy [14].

Measurable residual disease (MRD) is characterized by the presence of more than 1 CLL cell per 10,000 leukocytes [15]. The MRD status can be assessed using multicolor flow cytometry [16]. Notably, undetectable MRD serves as an independent predictor of prolonged PFS and OS [17]. Previously, MRD monitoring was limited to early periods post treatment. Importantly, it can also serve as an indicator of sustained remission during follow-ups.

In this study, we analyzed patients treated with FCR and the impact of severe myelosuppression as the risk factors. Moreover, we attempted to demonstrate the sustainability of CLL remission by checking the MRD.

## 2. Materials and Methods

### 2.1. Patients

We retrospectively reviewed the data of patients with CLL treated with the FCR chemoimmunotherapy regimen [18] in Seoul, Yeouido and Incheon St. Mary’s Hospital between 11 December 2009 and 9 January 2023. We included all consecutive patients with CLL (aged ≥ 18 years) who were diagnosed according to the International Workshop on CLL (iwCLL) [15]. Patients testing positive for human immunodeficiency virus or diagnosed with Richter’s syndrome were excluded.

### 2.2. Treatment, Response, and MRD Assessment

The FCR chemotherapy regimen comprised, as follows: 25 mg/m^2^/day fludarabine and 250 mg/m^2^/day cyclophosphamide on days 1–3; 375 mg/m^2^ rituximab on day 1 of cycle 1; and 500 mg/m^2^ rituximab on day 1 of cycles 2–6. We evaluated treatment response and disease progression based on iwCLL guidelines [15]. A response assessment was performed 5–6 weeks after the last completion of FCR. MRD assessments were performed using multiparameter flow cytometry according to the previously reported universal approach [19,20]. MRD analysis was conducted every 6 months until the last follow-up in patients who had sustained responses after the initiation of chemotherapy. In the setting of flow cytometric analysis, with at least 500,000 events acquired, MRD-negative status was defined as less than 0.01%.

IGHV mutation status was assessed using LymphoTrack IGH assay panels (InVivoScribe, San Diego, CA, USA). Amplification in 240 ng of high-quality DNA was performed with a single multiplex master mix containing primers; the library DNA was loaded onto a Miseq Reagent Kit, v2. LymphoTrack MRD software, v2.0.2 was used to analyze the FASTQ files [21]. *TP53* positivity status was evaluated at diagnosis through the detection of *TP53* mutations or by identifying a deletion of 17p.

Differences in the white blood cell (WBC) count, hemoglobin levels, and thrombocytopenia were assessed at baseline, 6, 9, and 12 months.

### 2.3. Statistical Analysis

All categorical variables were analyzed using the chi-square or Fisher’s exact test; continuous variables were analyzed using the Mann–Whitney *U* test and Student’s *t*-test.

OS was calculated from the date of treatment initiation to the date of death or the end of the follow-up period. PFS was assessed from initial chemotherapy to the date of death, treatment failure, or disease progression due to any cause. OS and PFS were analyzed using Kaplan–Meier curves and compared using the log-rank test. A Cox proportional hazards model was used to identify the factors significantly affecting OS in univariate and multivariate analyses.

For the cumulative incidence of relapse (CIR), relapse was calculated as an uncensored event, and death without relapse was considered a competing cause of relapse. Non-relapse mortality (NRM) was defined as mortality without relapse, with relapse considered as a competing risk factor. The CIR and NRM were analyzed using the cumulative incidence of competing events, Gray’s test for univariate analysis, and Fine–Gray proportional hazard regression for multivariate analysis. We used survival decision trees with the machine learning method for risk stratification. The minimal split trees were set to two. Variables that were analyzed in the algorithm were, as follows: the presence of bicytopenia (coexisting thrombocytopenia and anemia); age > 65 years at the initiation of chemotherapy; female sex; Binet B–C or Rai I–IV; β2 microglobulin > 3.5 mg/L; unmutated IGHV; positive *TP53* status; positivity for deletion 11; and complex karyotype [22].

All predictors with a *p*-value < 0.05 in the univariate analysis were included in the multivariate analysis. The reported *p*-values were two-sided, and statistical significance was set at *p* < 0.05. All statistical analyses were performed using R software (version 4.0.2; R Foundation for Statistical Computing, Vienna, Austria).

### 2.4. Ethics Statement

This study was approved by the Institutional Review Board and Ethics Committee of the Catholic Medical Center in the Republic of Korea (XC22RIDI0106). Patient consent was not required because of the retrospective nature of the study.

## 3. Results

### 3.1. Patient Characteristics

The baseline characteristics of the patients with CLL at diagnosis are presented in Table 1. We included 144 patients who received FCR. The results are reported after a median observation period of 6.2 (0.8–11.7) years. The patients had, as follows: a median age of 60 (27–82) years; 67.4% had Binet stage C disease; 7.6% had positive *TP53* status; and 78.3% had a mutated IGHV.

We defined the bicytopenia group as patients presenting with hemoglobin ≤ 11 g/dL and a platelet count ≤ 100,000/mm^3^, using a cut-off aligned with the Rai stage [23]. The bicytopenia group comprised 31.3% of the total patient cohort, whereas the non-bicytopenia group represented 68.7%. Patients older than 65 years at the initiation of chemotherapy, classified under Binet B–C or Rai I–IV stages, with serum β2 microglobulin levels > 3.5 mg/L, or those who did not complete six cycles of chemotherapy, were more prevalent in the bicytopenia group compared with the non-bicytopenia group.

### 3.2. Efficacy of FCR and Cumulative Incidence of Secondary Malignancies

The 5-year OS of all patients was 84.4% (95% confidence interval [CI], 78.3–90.6), and the 5-year PFS was 68.3%. According to the iwCLL response criteria, in the FCR group, the overall response rate was 84.7%, with complete remission (CR) accounting for 58.3% (*n* = 84), partial remission (PR) for 26.4% (*n* = 38), stable disease for 8.3% (*n* = 12), and progressive disease for 6.9% (*n* = 10).

We identified 15 cases of secondary malignancies, including, as follows: diffuse large B-cell lymphoma (*n* = 4); therapy-related myeloid neoplasms (*n* = 4); lung cancer (*n* = 3); multiple myeloma (*n* = 1); Burkitt lymphoma (*n* = 1); glioblastoma (*n* = 1); and colon cancer (*n* = 1). The median time to onset of secondary hematologic malignancies after the first dose of treatment was 48.6 months; the 5-year CIR was 5.5%.

### 3.3. Impact of the Bicytopenia Group: OS, PFS, CIR, TRM, and Prognostic Factors

The 5-year OS of the bicytopenia and non-bicytopenia groups was 74.6% (95% CI, 62.6–88.9) and 88.8% (95% CI, 82.4–95.7, *p* = 0.012), respectively. The 5-year PFS in the bicytopenia and non-bicytopenia was 52.5% and 75.6%, respectively (*p* < 0.001). The 5-year CIR in the bicytopenia and non-bicytopenia groups was 38.5% and 11.1%, respectively (*p* = 0.001). The 5-year NRM did not significantly differ between the two groups (Figure 1 and Table 2).

In the multivariate analysis, β2 microglobulin levels > 3.5 mg/L and positive *TP53* status were associated with worse OS. The presence of bicytopenia, a positive *TP53* status, and a complex karyotype adversely impacted PFS. Furthermore, the CIR was correlated with the presence of bicytopenia, a positive *TP53* status, deletion 11 positivity, anId a complex karyotype. An increase in NRM was associated with elevated levels of β2 microglobulin (Table 3).

Because the β2 microglobulin level (*n* = 102) and IGHV status (*n* = 46) were evaluated in the subgroup, a survival decision tree was employed for risk stratification. In analyzing the OS and PFS using the survival decision tree, the bicytopenia group and the non-bicytopenia group were divided at the first layer. Moreover, *TP53* status, both positive and negative, influenced the tree split (Figure 2A). Bicytopenia, combined with *TP53* positivity, was associated with inferior OS (Figure 2B).

Among the total cohorts, the international prognostic index for chronic lymphocytic leukemia (CLL-IPI) was applicable to 39 patients [24]. However, stratification by CLL-IPI did not show differences at any stage (Figure 2C). Conversely, risk stratification utilizing the survival decision tree revealed poorer OS outcomes in the cytopenia and *TP53*-positive group compared with the non-bicytopenia group.

### 3.4. Suppression in Platelets, Hemoglobin, WBCs, and MRD

To evaluate the sustained suppression of the complete blood count, we assessed the differences in platelet, hemoglobin, and WBC counts at initiation and 6, 9, and 12 months after. Delayed recovery from thrombocytopenia and anemia was observed in the bicytopenia group, compared with the non-bicytopenia group, until 12 months after chemotherapy (Figure 3A,B). However, the recovery of WBC and MRD was similar between the two groups (Figure 3B,C). MRD was assessed at the last follow-up for 72 patients who achieved CR or PR. MRD negativity was sustained for a median of 5 (1–14) years and a maximum of 14 years (Figure 3D).

## 4. Discussion

We assessed the impact of cytopenia on treatment outcomes with FCR therapy. Specifically, we identified a delayed recovery in anemia and platelets and its link with the inability to complete full cycles of chemoimmunotherapy. These bone marrow-suppressive effects ultimately contributed to increased relapse rates. In the Republic of Korea, despite patients with CLL constituting a small percentage of all patients with mature B-cell lymphoma, the disease exhibits a relatively aggressive pattern. The 5-year OS rate ranges from 71% to 79.3%, which is below the rates observed in Western countries [25,26]. Following the adoption of FCR and the establishment of a standard therapy, recent data indicate an improvement to 84.4% in our dataset. Herling et al. reported that early FCR application in patients with Binet A and high-risk factors, such as unfavorable cytogenetics, did not significantly enhance survival compared to the watch-and-wait strategy [27]. Thus, patients with anemia and thrombocytopenia, corresponding to Binet C or Rai III–IV, are prime candidates for this treatment. Regarding hematologic toxicity, grades 3 and 4 thrombocytopenia occurred in 6% to 7% of patients treated with FCR [3,7]. However, 43% of cases exhibited delayed recovery, which must be distinguished from treatment-related myeloid neoplasm [28,29]. In this study, patients with bicytopenia undergoing FCR therapy demonstrated delayed recovery in thrombocytopenia and anemia, compared to their counterparts, over the 12 months following the initiation of chemotherapy. This trend was more pronounced among FCR non-responders than FCR responders, complicating the transition to second-line therapies, such as ibrutinib, which poses a bleeding risk in patients with severe thrombocytopenia [30]. Our study suggested that both insufficient chemoimmunotherapy cycles and the delayed application of Bruton’s tyrosine kinase inhibitor are associated with increased relapse rates and progression-related mortality.

Thompson et al. reported long-sustained remission in CLL following FCR treatment. The median PFS was approximately 14.6 years in the IGHV mutation group. However, our study did not demonstrate a difference in PFS based on IGHV mutational status. This discrepancy might be attributable to the short duration of the follow-up; moreover, a higher prevalence of IGVH mutations at 78.3%, compared with the 61% previously reported, could influence outcomes [13]. A high incidence of IGVH mutations among Koreans suggests that FCR is a significant option for frontline therapy. This study verified that the therapy could achieve extended MRD negativity. Furthermore, in comparison with new agents—with durations of 12–27 months for venetoclax treatment and 27–36 months for ibrutinib treatment [9,10]—6 months of FCR treatment could be more cost-effective when selecting a good responder. We suggest that, for the high-risk group, particularly young patients susceptible to developing profound cytopenia due to leukemia refractoriness, consideration should be given to alternatives to FCR for frontline use [8,31]. This study involved different patients compared to a previous cohort discussed by Kim et al. [14]. We aimed to assess the impact of FCR therapy on MRD outcomes and the high incidence of IGHV-mutated status in the Republic of Korea. Within the health insurance system, our findings aim to help identify patients who require additional caution or alternative therapeutic recommendations based on their risk profiles.

Our study has some limitations. First, the relatively small number of patients and the study’s retrospective nature preclude definitive conclusions. Additionally, owing to the different sizes of the comparison groups, our results require careful interpretation. However, given the rarity of CLL in the Republic of Korea, real-world data may support the selection of FCR among various frontline therapies. The higher genetic proportions of IGHV mutation status, relative to Western countries, reinforce this suggestion. Second, the existing risk stratification system shows discrepancies, possibly due to the brief follow-up period and the small number of patients in each group; further research is needed to confirm the impact of risk stratification. Nevertheless, our study aimed to highlight a significant infection risk and the potential challenges in transitioning to subsequent therapies. In the Republic of Korea, the prognosis of the disease has historically been less favorable, compared with Western countries, influenced by factors such as lymphocyte doubling time [32]. Nonetheless, there has been a gradual enhancement in the application of first-line FCR therapy, including strategies to minimize infection risks by shortening the nadir phase through prophylaxis with a long-acting granulocyte colony-stimulating factor [33].

## 5. Conclusions

Patients with bicytopenia, particularly those experiencing prolonged cytopenia after FCR, exhibit poor survival outcomes. The selection of a first-line therapy must be cautiously performed, considering the status of bicytopenia and *TP53* mutations in addition to the IGHV mutational status. This study illuminates the challenges associated with cytopenia following FCR and the complexities involved in transitioning to alternative treatments. A tailored selection can benefit patients who have undergone FCR therapy, particularly those with a brief treatment period and prolonged MRD negativity.

## Figures and Tables

**Figure 1 biomedicines-13-00194-f001:**
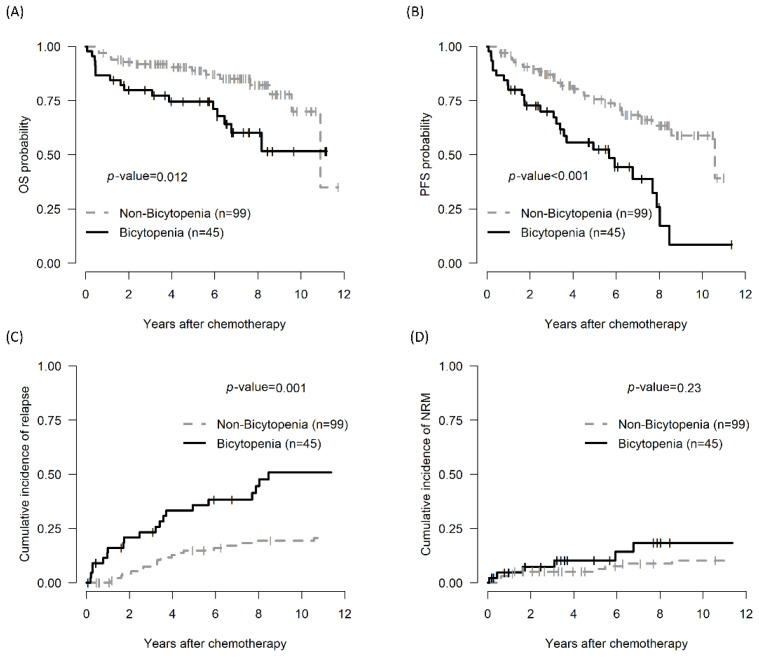
Comparison of treatment outcomes between the bicytopenia and non-bicytopenia groups among patients with CLL: OS (**A**); PFS (**B**); CIR (**C**); and TRM (**D**). Bicytopenia: coexistence of hemoglobulin ≤ 11 g/dL and platelets ≤ 100,000/mm^3^; CLL: chronic lymphocytic leukemia; OS: overall survival; PFS: progression-free survival; NRM: non-relapse mortality.

**Figure 2 biomedicines-13-00194-f002:**
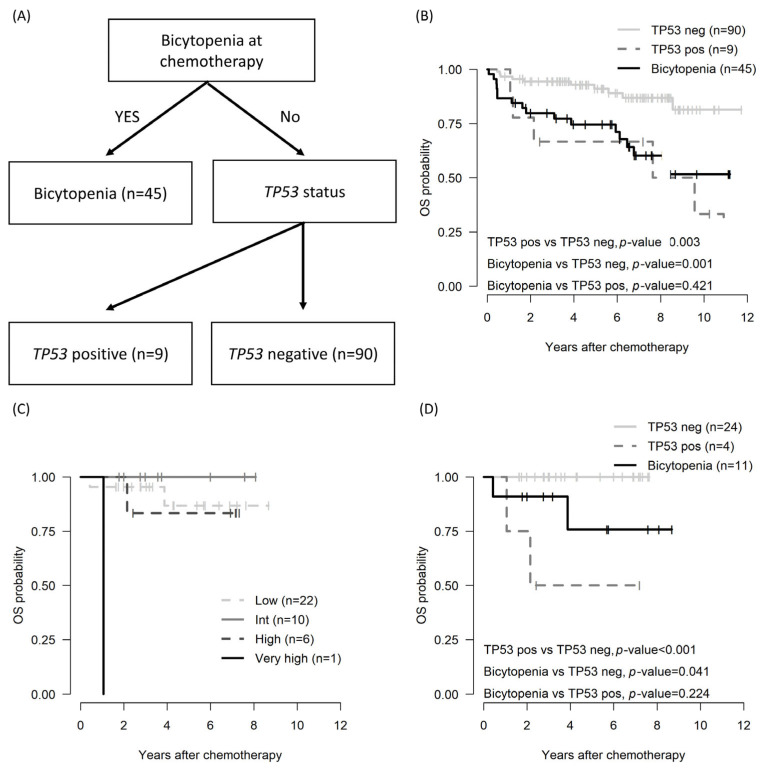
OS by risk stratification in patients with CLL: decision tree based on survival decision tree algorithm (**A**); OS across the entire cohort utilizing the survival decision tree model (**B**), OS in subgroup patients assessed by CLL-IPI (**C**); and OS in subgroup patients determined through the survival decision tree model (**D**). CLL, chronic lymphocytic leukemia; CLL-IPI: an international prognostic index for patients with chronic lymphocytic leukemia; pos: positive; neg: negative; OS: overall survival.

**Figure 3 biomedicines-13-00194-f003:**
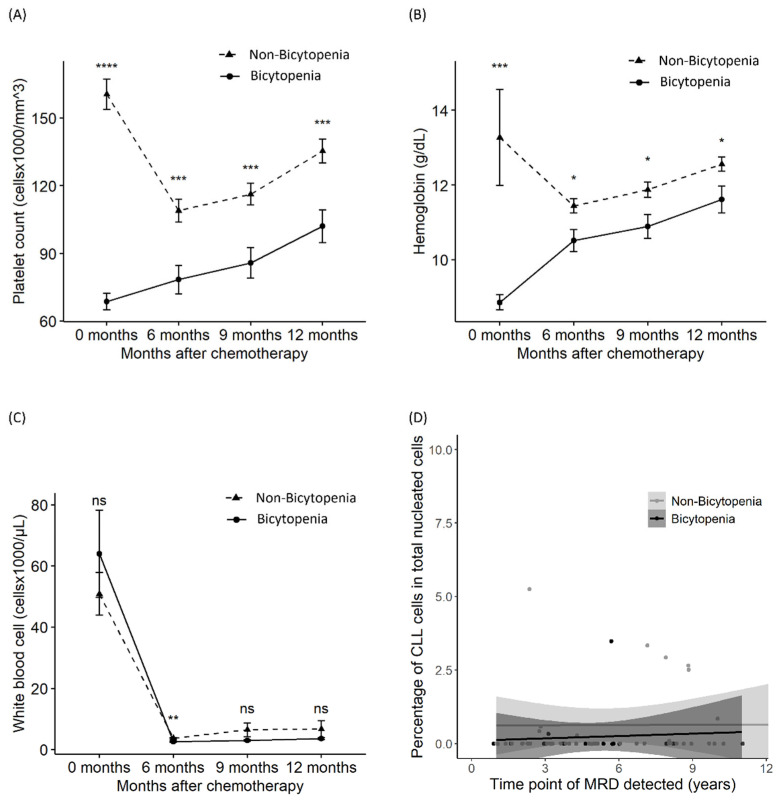
Differences in platelet count, hemoglobin level, white blood cell count, and MRD between the bicytopenia and non-bicytopenia groups. After chemotherapy initiation, comparisons were made between the two groups in terms of: platelet count (**A**); hemoglobin level (**B**); absolute lymphocyte count (**C**); and MRD—percentage of abnormal cells of total nucleated elements (**D**). Circles and triangles represent the mean values for each group; bars denote the confidence intervals determined by the t-test. FCR: fludarabine, cyclophosphamide, and rituximab; MRD: measurable residual disease. * *p* < 0.05, ** *p* < 0.01, *** *p* < 0.001, and **** *p* < 0.0001; ns: non-significant.

**Table 1 biomedicines-13-00194-t001:** Characteristics of patients with chronic lymphocytic leukemia.

	Total	Non-Bicytopenia (*n* = 99)	Bicytopenia (*n* = 45)	*p*-Value
Age > 65 years at chemotherapy, N (%)	44 (30.6)	23 (23.2)	21 (46.7)	0.008
Sex, female, N (%)	53 (36.8)	37 (37.4)	16 (35.6)	0.981
Binet stage, N (%)				<0.001
−A	14 (9.7)	14 (14.1)	0 (0.0)	
−B	33 (22.9)	33 (33.3)	0 (0.0)	
−C	97 (67.4)	52 (52.5)	45 (100.0)	
Rai points, N (%)				<0.001
0	11 (7.6)	11 (11.1)	0 (0.0)	
−1	23 (16.0)	23 (23.2)	0 (0.0)	
−2	13 (9.0)	13 (13.1)	0 (0.0)	
−3	33 (22.9)	33 (33.3)	0 (0.0)	
−4	64 (44.4)	19 (19.2)	45 (100.0)	
Binet B–C or Rai I–IV, N (%)	97 (67.4)	52 (52.5)	45 (100.0)	<0.001
Serum β2 microglobulin, N (%)				0.001
>3.5 mg/L	20 (19.6)	8 (11.0)	12 (41.4)	
≤3.5 mg/L	82 (80.4)	65 (89.0)	17 (58.6)	
IGHV mutational status, N (%)				0.23
Unmutated	10 (21.7)	9 (28.1)	1 (7.1)	
Mutated	36 (78.3)	23 (71.9)	13 (92.9)	
*TP53* status positivity, N (%)	11 (7.6)	9 (9.1)	2 (4.4)	0.526
Deletion 11 positivity, N (%)	15 (10.4)	8 (8.1)	7 (15.6)	0.286
Complex karyotype, N (%)	22 (15.3)	16 (16.2)	6 (13.3)	0.851
Failure to complete six cycles of chemotherapy, N (%)	32 (22.2)	16 (16.2)	16 (35.6)	0.017

*TP53* status positivity, deletion 17p (FISH), and/or *TP53* mutation (sequencing). IGHV, immunoglobulin heavy-chain variable.

**Table 2 biomedicines-13-00194-t002:** Univariate analyses of survival outcomes.

	Overall Survival		Progression-Free Survival	
	HR (95% CI)	*p*	HR (95% CI)	*p*
Age > 65 years vs. ≤65 years	1.38 (0.66, 2.87)	0.389	1.17 (0.66, 2.09)	0.586
Sex, female vs. male	0.84 (0.40, 1.75)	0.641	0.84 (0.48, 1.47)	0.531
Bicytopenia vs. non-bicytopenia	2.38 (1.19, 4.79)	0.015	2.81 (1.64, 4.83)	<0.001
Binet B–C or Rai I–IV vs the other	2.46 (1.01, 5.99)	0.047	1.56 (0.86, 2.83)	0.147
β2 microglobulin, >3.5 vs. ≤3.5	3.49 (1.51, 8.05)	0.003	2.47 (1.28, 4.76)	0.007
IGHV, unmutated vs. mutated	1.04 (0.11, 10.1)	0.97	0.82 (0.21, 3.15)	0.775
*TP53* status, positive vs. negative	3.03 (1.30, 7.08)	0.01	2.71 (1.32, 5.59)	0.007
Deletion 11, positive vs. negative	1.11 (0.39, 3.18)	0.846	2.23 (1.08, 4.60)	0.031
Complex karyotype vs. the other	0.86 (0.30, 2.45)	0.775	2.16 (1.13, 4.14)	0.02
	Cumulative incidence of relapse		Cumulative incidence of non-relapse	
	HR (95% CI)	*p*	HR (95% CI)	*p*
Age > 65 years vs. ≤65	3.06 (1.48, 6.36)	0.77	0.39 (0.05, 2.99)	0.71
Sex, female vs. male	1.02 (0.53, 1.95)	0.96	0.58 (0.19, 1.79)	0.34
Bicytopenia vs. non-bicytopenia	2.87 (1.55, 5.30)	0.001	1.52(0.54, 4.23)	0.43
Binet B–C or Rai I–IV vs. the other	1.43 (0.72, 2.82)	0.31	1.45 (0.48, 4.41)	0.52
β2 microglobulin, >3.5 vs. ≤3.5	1.3 (0.56, 3.02)	0.55	4.13 (1.47, 11.57)	0.007
IGHV, unmutated vs. mutated	0.49 (0.11, 2.10)	0.33	NA	NA
*TP53* status, positive vs. negative	2.64 (1.33, 5.25)	0.006	1.73 (0.37, 8.01)	0.48
Deletion 11, positive vs. negative	3.71 (1.93, 7.16)	<0.001	NA	NA
Complex karyotype vs. the other	3.06 (1.48, 6.35)	0.003	0.39 (0.05, 2.99)	0.37

HR, hazard ratio; *p*, *p*-value; *TP53* status positivity, deletion 17p (FISH), and/or *TP53* mutation (sequencing); NA, not applicable.

**Table 3 biomedicines-13-00194-t003:** Multivariate analyses of survival outcomes.

	Overall Survival		Progression-Free Survival		Cumulative Incidence of Relapse	
	HR (95% CI)	*p*	HR (95% CI)	*p*	HR (95% CI)	*p*
Bicytopenia vs. non-bicytopenia	2.43 (0.98, 6.05)	0.056	3.67 (1.70, 7.90)	<0.001	3.41 (1.76, 6.61)	<0.001
β2 microglobulin, >3.5 vs. ≤3.5	2.92 (1.15, 7.43)	0.025	1.69 (0.82, 3.49)	0.158		
*TP53* status, positive vs. negative	3.50 (1.11, 11.1)	0.033	2.61 (1.03, 6.60)	0.042	3.07 (1.37, 6.85)	0.006
Deletion 11, positive vs. negative			1.07 (0.41, 2.80)	0.89	2.42 (1.26, 4.67)	0.008
Complex karyotype vs. the other			3.02 (1.09, 8.37)	0.034	2.38 (1.04, 5.44)	0.04

HR, hazard ratio; *TP53* status positivity, deletion 17p (FISH), and/or *TP53* mutation (sequencing).

## Data Availability

The data presented in this study are available on request from the corresponding author. The data are not publicly available owing to privacy and ethical reasons.

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
