# Peer review of "Inferior Outcomes of Fludarabine–Cyclophosphamide–Rituximab Chemotherapy in Korean Chronic Lymphocytic Leukemia Patients with Concurrent Thrombocytopenia and Anemia"

_biomedicines, 2025, doi:10.3390/biomedicines13010194_

Round 1

Reviewer 1 Report

Comments and Suggestions for Authors

The manuscript submitted by Kim et al., entitled "Inferior outcomes of concurrent thrombocytopenia and anemia in patients with chronic lymphocytic leukemia with fludarabine–cyclophosphamide–rituximab chemotherapy," is well-written and informative. Given the very limited on treatment outcomes CLL using FRP from Asia, it worth considering for publication in the journal Biomedicines. However, I have the following minor comments for the authors to enhance the clarity of their manuscript:

1.     The title lacks clarity. It should be modified to "Inferior outcomes of fludarabine–cyclophosphamide–rituximab chemotherapy in Korean chronic lymphocytic leukemia patients with concurrent thrombocytopenia and anemia" or "Inferior outcomes of fludarabine–cyclophosphamide–rituximab chemotherapy in Korean chronic lymphocytic leukemia patients with concurrent bicytopenia and TP53 status-positive."

2.     The definition of the control group is unclear. Typically, a control group refers to patients who received no treatment or a different treatment regimen. According to the authors, the control group refers to CLL patients without concurrent bicytopenia and/or TP53 status-positive but treated with FCR like the case group. This should be succinctly and clearly stated.

3.     The number of CLL patients in the bicytopenia group (n=44) and the control group (n=99) is significantly different. This small size of the bicytopenia group might compromise the findings. Although the authors have mentioned this as a limitation, precautions should be considered when extrapolating findings from this study.

4.      Further risk stratification analysis of FCR treatment outcomes for CLL patients was severely compromised by the extremely small size of the case group. This limitation should be clearly discussed.

The authors should clarify whether the same CLL patients were used in their previous multicenter study entitled "Treatment pattern of chronic lymphocytic leukemia/small lymphocytic lymphoma in Korea: a multicenter retrospective study (KCSG LY20-06)." If so, they should highlight what new information this study provides that was not reported in their previous study findings. Both studies employed a retrospective design with similar sample sizes, so it is important to clarify the 

Author Response

Reviewer 1 Comments for the Author:

  1. The title lacks clarity. It should be modified to "Inferior outcomes of fludarabine–cyclophosphamide–rituximab chemotherapy in Korean chronic lymphocytic leukemia patients with concurrent thrombocytopenia and anemia" or "Inferior outcomes of fludarabine–cyclophosphamide–rituximab chemotherapy in Korean chronic lymphocytic leukemia patients with concurrent bicytopenia and TP53 status-positive."

-> We would like to thank you for evaluating our manuscript and for the insightful comments. To avoid potential confusion for readers, we have revised the title and hope that this adjustment fulfills the intent of the original comment.

(Before revision, Title) Line 2

Inferior outcomes of concurrent thrombocytopenia and anemia in patients with chronic lymphocytic leukemia with fludara-bine–cyclophosphamide–rituximab chemotherapy

(After revision, Title) Line 5

Inferior outcomes of concurrent thrombocytopenia and anemia in patients with chronic lymphocytic leukemia with fludara-bine–cyclophosphamide–rituximab chemotherapy

Inferior outcomes of fludarabine–cyclophosphamide–rituximab chemotherapy in Korean chronic lymphocytic leukemia patients with concurrent thrombocytopenia and anemia

  1. The definition of the control group is unclear. Typically, a control group refers to patients who received no treatment or a different treatment regimen. According to the authors, the control group refers to CLL patients without concurrent bicytopenia and/or TP53 status-positive but treated with FCR like the case group. This should be succinctly and clearly stated.

-> We apologize for the confusion and thank you for highlighting the ambiguity in this part. We have opted to use the term “non-bicytopenia group” instead of “control group” to avoid confusion among readers.

  1. The number of CLL patients in the bicytopenia group (n=44) and the control group (n=99) is significantly different. This small size of the bicytopenia group might compromise the findings. Although the authors have mentioned this as a limitation, precautions should be considered when extrapolating findings from this study.

à Thank you for highlighting the potential impact of the unequal group sizes on our findings. We agree that the smaller group may affect the generalizability of the results. Consequently, we have now included an explicit statement in the manuscript to emphasize this limitation.

(Before revision, Table 2) Line 237

Our study has some limitations. First, the relatively small number of patients and the study's retrospective nature preclude definitive conclusions.

(After revision, Table 2) Lines 255–257

Our study has some limitations. First, the relatively small number of patients and the study's retrospective nature preclude definitive conclusions. Additionally, owing to the different sizes of the comparison groups, our results require careful interpretation.

  1. Further risk stratification analysis of FCR treatment outcomes for CLL patients was severely compromised by the extremely small size of the case group. This limitation should be clearly discussed.

-> The noted discordance is ascribed to our center's strategy of using ProMACE chemotherapy as the first-line therapy for extranodal NK lymphoma. We have addressed this limitation in the manuscript.

(Before revision, Discussion) Line 241

Second, the existing risk stratification system shows discrepancies, possibly attributable to the brief follow-up period.

(After revision, Discussion) Lines 260–263

Second, the existing risk stratification system shows discrepancies, possibly due to the brief follow-up period and the small number of patients in each group; further research is needed to confirm the impact of risk stratification.

  1. The authors should clarify whether the same CLL patients were used in their previous multicenter study entitled "Treatment pattern of chronic lymphocytic leukemia/small lymphocytic lymphoma in Korea: a multicenter retrospective study (KCSG LY20-06)." If so, they should highlight what new information this study provides that was not reported in their previous study findings. Both studies employed a retrospective design with similar sample sizes, so it is important to clarify

-> Our study cohort differs from the previous one, a multicenter retrospective study (KCSG LY20-06). The previous research lacked consistency in evaluating the impact of MRD or IGHV mutational status, as well as time-dependent variables, such as hemoglobulin and platelet. Therefore, we have also mentioned these differences in the previous report.

(Before revision, Discussion) Line 234

(…) We suggest that for the high-risk group, particularly younger patients susceptible to de-veloping profound cytopenia from leukemia refractoriness, consideration should be given to alternatives to FCR for frontline use [8, 32].

(After revision, Discussion) Lines 247–254

(…)We suggest that, for the high-risk group, particularly young patients susceptible to developing profound cytopenia due to leukemia refractoriness, consideration should be given to alternatives to FCR for frontline use [8,31]. This study involved different patients, compared with a previous cohort discussed by Kim et al. [14]. We aimed to assess the impact of FCR therapy on MRD outcomes and the high incidence of IGHV-mutated status in Korea. Within the health insurance system, our findings aim to help identify patients who require additional caution or alternative therapeutic recommendations based on their risk profile.

Reviewer 2 Report

Comments and Suggestions for Authors

In this study, the authors investigated the effect of cytopenia on the OS and PFS of Korean patients with CLL treated with chemoimmunotherapy. They found that bicytopenia (thrombocytopenia and anemia) is an adverse risk factor.

This result is not new and nor the fact that CLL patients with TP53 problems have a poor prognosis.

Comments and questions:

 1. How were the flow cytometric measurements performed? How many events were measured from each sample?

2. What does "TP53 positive status" mean? It wasn't defined in the manuscript.

3. Please improve the format of the Table 2. Make the separation better on its part.

4. The Figure 3D part is not clear. What is on the vertical axes? What is the unit of this? There is no legend for this part of the figure.

Author Response

Reviewer 2 Comments for the Author:

  1. How were the flow cytometric measurements performed? How many events were measured from each sample?

-> Our centers evaluated the flow cytometric measurements after complete remission, and we examined the MRD every 6 months. However, the ranges varied; therefore, we have only described the last follow-up outcomes.

(Before revision, Material and Methods) Line 80

MRD analysis was conducted during the last follow-up in patients who had sustained responses after the initiation of chemotherapy.

(After revision, Material and Methods) Line 86

MRD analysis was conducted every 6 months until the last follow-up in patients who had sustained responses after the initiation of chemotherapy.

  1. What does "TP53 positive status" mean? It wasn't defined in the manuscript.

-> Thank you for noting the potential ambiguity surrounding “TP53 positivity status.” We agree that further clarification is necessary. Therefore, we have now included a clear definition in the manuscript to explain what we mean by “TP53 positivity status.”

(Before revision, Method) Line 85

(…) The LymphoTrack MRD software v2.0.2 was used to analyze the FASTQ files [22].

(After revision, Discussion) Lines 91–93

(…) The LymphoTrack MRD software v2.0.2 was used to analyze the FASTQ files [21]. TP53 positivity status was evaluated at diagnosis through detection of TP53 mutations or by identifying a deletion of 17p.

  1. Please improve the format of the Table 2. Make the separation better on its part.

->To improve the clarity of Table 2, we have followed your suggestion and separated the univariate and multivariate analyses into a new table, Table 3.

(Before revision, Discussion) Line 150, Line 161

(Line 155) The 5-year NRM did not significantly differ between the two groups (Figure 1).

(Line 161) An increase in NRM was associated with elevated levels of β2 microglobulin (Table 2).

(After revision, Discussion) Lines 159 and 171

(Lines 158–159) The 5-year NRM did not significantly differ between the two groups (Figure 1 and Table 2).

(Line 170–171) An increase in NRM was associated with elevated levels of β2 microglobulin (Table 3).

     4. The Figure 3D part is not clear. What is on the vertical axes? What is the unit of this? There is no legend for this part of the figure.

-> We apologize for the confusion and thank you for highlighting the ambiguity in this part. We have revised the figure to include clearer labeling of the vertical axis and the appropriate units, as well as a more detailed legend explaining the data depicted.

(Before revision, Figure 3) Line 192

Figure 3. Differences in platelet count, hemoglobin level, white blood cell count, and MRD between the bicytopenia and control groups. After chemotherapy initiation, the two groups were compared regarding platelet count (A), hemoglobin level (B), absolute lym-phocyte count, and MRD (C).

(After revision, Figure 3) Lines 205–209

Figure 3. Differences in platelet count, hemoglobin level, white blood cell count, and MRD between the bicytopenia and control non-bicytopenia groups. After chemotherapy initiation, comparisons were made between the two groups in terms of platelet count (A), hemoglobin level (B), absolute lymphocyte count (C), and MRD: percentage of abnormal cells of total nucleated elements (C) (D).

Round 2

Reviewer 2 Report

Comments and Suggestions for Authors

The sensitivity of flow cytometric measurement and the level of MRD depend on the number of measured events. Please define the range of measured event numbers to see the level of MRD. In the case of 100,000 measured events, the MRD level is 0.05%, while at 500,000 events, it is 0.01%. According to the cited publications, this level is considered MRD-negative. If the measurements did not reach this level, then the results and the conclusion should be revised.

Author Response

Reviewer 2 (Round 2) Comments for the Author:

  1. The sensitivity of flow cytometric measurement and the level of MRD depend on the number of measured events. Please define the range of measured event numbers to see the level of MRD. In the case of 100,000 measured events, the MRD level is 0.05%, while at 500,000 events, it is 0.01%. According to the cited publications, this level is considered MRD-negative. If the measurements did not reach this level, then the results and the conclusion should be revised.?

-> We understand your concerns regarding the sensitivity of MRD. All MRDs were evaluated with at least 500,000 events, and undetectable MRD is defined as <0.01%. This clarification has been added to the Methods section.

(Before revision, Material and Methods) Line 83

MRD analysis was conducted every 6 months until the last follow-up in patients who had sustained responses after the initiation of chemotherapy.

(After revision, Material and Methods) Line 83

MRD analysis was conducted every 6 months until the last follow-up in patients who had sustained responses after the initiation of chemotherapy. In the setting of flow cytometric analysis, with at least 500,000 events acquired, MRD-negative status is defined as less than 0.01%.

Round 3

Reviewer 2 Report

Comments and Suggestions for Authors

Now, it is well described.